# Canine Leishmaniosis Associated with Acute Pleural Effusion and Sudden Death in a Dog

**DOI:** 10.3390/vetsci11060254

**Published:** 2024-06-04

**Authors:** Maria Caroline Pereira Brito, Maria de Fátima Sousa, Rubia Avlade Guedes Sampaio, Markyson Tavares Linhares, Lourdes Fernandez Riquelme, Wellida Karinne Lacerda, Ricardo Barbosa Lucena

**Affiliations:** 1Graduate Program in Animal Science, Universidade Federal da Paraiba, Areia 58397-000, PB, Brazil; clinica4patas@gmail.com (M.C.P.B.); exames@casadoscriadores.com.br (M.d.F.S.); lfernandez@vet.una.py (L.F.R.); wklc@academico.ufpb.br (W.K.L.); 2Clinic 4Patas, João Pessoa 58030-330, PB, Brazil; 3Casa dos Criadores, João Pessoa 58040-330, PB, Brazil; 4Graduate Program in Animal Science and Health, Universidade Federal de Campina Grande, Patos 58708-110, PB, Brazil; rubia_avlade@estudante.ufcg.edu.br; 5Undergraduate Course in Veterinary Medicine, Center for Agricultural Sciences, Universidade Federal da Paraiba, Areia 58397-000, PB, Brazil; mtl@academico.ufpb.br; 6Laboratory of Veterinary Pathology, Universidade Federal da Paraiba, Areia 58397-000, PB, Brazil

**Keywords:** hound, *Leishmania* sp., pleural disease, zoonosis

## Abstract

**Simple Summary:**

Pleural effusion is a condition characterized by excessive fluid accumulation in the pleural spaces. This condition is a consequence of an underlying disease and may be caused by many factors, such as kidney, liver, and heart diseases; systemic inflammation; and neoplasms. It can occur acutely or chronically. We diagnosed the occurrence of acute pleural effusion as a result of leishmaniasis and necrotizing lymphadenitis in a dog. The diagnosis was confirmed through necropsy, as well as the identification of *Leishmania* amastigotes in the pleural fluid, lymph nodes, and other organs. In humans, the occurrence of acute pleural effusion associated with leishmaniasis has also been described, however, to the authors’ knowledge, this association has not yet been described in dogs. This study describes the first report of a dog with leishmaniasis associated with acute pleural effusion that died suddenly.

**Abstract:**

A two-year-old female crossbreed dog, previously a stray with no known owner, was adopted and subsequently spayed. The dog exhibited weight loss over a period of two months and died suddenly during a leashed walk. Upon necropsy, enlargement of the submandibular, prescapular, and popliteal lymph nodes was noted. The intrathoracic cavity contained a substantial volume of yellowish-white fluid. Lymph nodes in the mediastinal and ventral thoracic centers were also enlarged, hemorrhagic, and friable. Microscopic examination revealed significant architectural changes in the lymph nodes, characterized by a pronounced cellular infiltrate consisting of lymphocytes and histiocytes, along with macrophages containing intracytoplasmic *Leishmania* amastigotes. Immunohistochemical analysis of the lymph nodes confirmed positive staining for *Leishmania* amastigotes. This case represents the first report of canine leishmaniasis associated with acute pleural effusion and sudden death.

## 1. Introduction

Pleural effusion is a condition in which excessive accumulation of fluid (transudate, exudate, chylous, lymph, or blood) occurs in the pleural space. This intrathoracic accumulation may occur when capillary hydrostatic pressure or permeability is increased, intravascular oncotic pressure is decreased, or lymphatic drainage is impeded [1,2]. Therefore, pleural effusion is not a disease entity in itself, but a sign of an underlying disease [1].

In dogs, pleural effusion may be due to traumatic causes; congenital diseases; primary chest neoplasms and secondary thoracic metastases; abdominal neoplasms; and inflammatory, cardiac, or infectious diseases, such as vascular diseases and rodenticide poisoning [3,4]. Cytological evaluations of the pleural effusions may suggest the pathogenesis of a disease [5].

Leishmaniosis is a chronic infectious disease primarily caused by *Leishmania infantum*, characterized by a wide range of symptoms, including respiratory issues such as coughing and nasal discharge [6]. Clinical lung involvement in leishmaniosis is considered rare [7]. These respiratory symptoms often stem from the systemic spread of the infection to the lungs, leading to secondary respiratory complications [6].

The accumulation of intrathoracic fluids as a complication of leishmaniosis has rarely been described in humans [7]. Only two cases of acute effusion have been reported in dogs affected by *Leishmania* sp. involving pericardial effusion [8,9]. In these cases, the citological examinations showed the parasites were found in the cytoplasm of some macrophages.

To the authors’ knowledge, acute canine pleural effusion as a complication of this disease has not been reported. Therefore, the aim of this study is to describe a case of acute pleural effusion in a dog with leishmaniosis.

## 2. Case Presentation

### 2.1. Clinical Features

A 2-year-old intact female crossbreed dog (10.6 kg), initially a stray that was later adopted, underwent ovariohysterectomy surgery without showing clinical signs. Over the following two months, the dog exhibited progressive weight loss, onychogryphosis, and mild nasal hyperkeratosis. Given these signs, the animal was subjected to serological testing to confirm the suspicion of leishmaniosis. The EIE-LVC kit produced by BioManguinhos/FIOCRUZ was tested according to Laurenti et al. [10]; however, the sample obtained a negative result. Nevertheless, before further tests could be conducted, during a walk, the dog exhibited signs of severe dyspnea and died suddenly. Subsequently, the dog was submitted for necropsy.

### 2.2. Gross Pathology and Cytology

During the macroscopic evaluation, it was observed that the dog had multifocal areas of alopecia on the extremities of all limbs, as well as hyperkeratosis of the nasal epithelium and all four pads. The submandibular, prescapular, and popliteal lymph nodes were enlarged. Upon opening the chest cavity, a large amount of yellowish fluid was observed (Figure 1A), resulting in significant compression of the lungs and pulmonary atelectasis. The sternal lymph nodes were enlarged, hemorrhagic, and friable (Figure 1B). The abdominal cavity contained a small amount of reddish fluid and a moderately enlarged liver with rounded edges.

During the necropsy, a sample of the intrathoracic fluid was centrifuged, and the sediment was used for the preparation of cytological smears, and was stained using the Romanowsky technique [11]. The cytological analysis of the pleural effusion revealed a significant presence of lymphocytes, plasma cells, and macrophages. Numerous macrophages contained intracytoplasmic Leishmania amastigotes. These structures were also observed extracellularly (Figure 1C). Given the turbid coloration and its intense cellularity, the thoracic fluid was classified as an exudate. Cytological analysis of the thoracic, mandibular, and popliteal lymph nodes, as well as the spleen, bone marrow, and bloods smear of the ear tip (Figure 1D) also revealed the presence of numerous Leishmania amastigotes within the cytoplasm of macrophages. Fragments of all internal organs, bone marrow, brain, spinal cord, and skin were collected and fixed in 10% buffered formalin solution. After fixation, the samples were sectioned, routinely processed in solutions of alcohols and xylenes, and stained with hematoxylin and eosin.

### 2.3. Histopathology

Microscopically, pulmonary atelectasis was observed (Figure 2) and changes in the architecture of the lymph nodes were observed, resulting from a marked cellular infiltrate composed of lymphocytes and histiocytes, associated with macrophages with intracytoplasmic *Leishmania* amastigotes and neutrophils (Figure 3A,B), as well as areas of hemorrhage and necrosis. The bone marrow showed inflammatory infiltrate composed of foamy macrophages, plasma cells. *Leishmania* amastigotes were found in the cytoplasm of macrophages. Microscopic evaluation of the skin revealed marked diffuse inflammatory infiltrate of lymphocytes, plasma cells, and macrophages with intracytoplasmic structures compatible with *Leishmania* amastigotes.

The liver sinusoids were filled with many red blood cells (congestion) and with the presence of many macrophages containing amastigote forms of *Leishmania* in the cytoplasm. In the renal interstitium, there were multiple discrete foci of lymphoplasmacytic inflammatory infiltrate. Some glomeruli were thickened and there was proliferation of mesangial cells; additionally, other glomeruli were sclerotic (early stage membranoproliferative glomerulonephritis). No lesions were observed in the other organs, including the heart and nervous system.

### 2.4. Immunohistochemistry

To confirm the diagnosis of Leishmaniosis associated with the lesions of this dog, an immunohistochemical examination of the lymph node was performed using a polyclonal anti-*Leishmania* antibody, according to Laurenti et al. [9]. Positive and intense staining of *Leishmania* amastigotes was observed, mainly in the cytoplasm of macrophages (Figure 3C,D).

## 3. Discussion

To the authors’ knowledge, the dog described in this study is the first case of canine leishmaniosis associated with acute pleural effusion and sudden death in a dog reported in the veterinary literature. The cause of death of this animal was respiratory failure due to a large amount of pleural fluid in the thorax, which compressed the lungs, causing atelectasis and preventing gas exchange. In some cases, fluid accumulation can be severe and cause tamponade, thus compromising visceral function [2]. Aspects of pleural effusion in this animal are similar to reports described in the literature, which characterize it as the accumulation of fluid in the chest. Pleural effusion represents a disturbance of this equilibrium, because of both increased production and decreased resorption [1].

The severe necrohemorrhagic lymphadenitis identified in the sternal lymph nodes indicates that this lesion has a direct relationship with the occurrence of acute effusion. Effusions may develop in local (altered organ) or systemic diseases, and the pathophysiology involved in fluid accumulation varies by etiology [12]. In dogs, the main causes of pleural effusions are heart disease, neoplasms, congenital disorders, and infections of the thoracic cavity [2,3]. However, in this case, no cardiac alterations were found that would justify an increase in hydrostatic pressure. Also, there were no lesions compatible with edema related to hypoproteinemia, as no hepatic lesions were found.

The renal injuries observed in the dog in this study ranged from mild to moderate in severity. Notably, there was an absence of other findings typically associated with hypoproteinemia in dogs with uremia, such as ascites and pulmonary edema [5]. However, despite the lack of blood tests performed on the patient, given the renal injuries observed, although of low severity, the liquid present in the thoracic cavity of this dog could also be of the transudate type, due to hypoproteinemia. Therefore, the liquid may be a mixture of exudate and transudate, as there was marked cellularity in the sediment, but a hydrothorax evidenced by hypoproteinemia cannot be ruled out [5]. In the present case, pleural effusion was the most relevant clinical abnormality. In cytology, macrophages with intracytoplasmic Leishmania amastigotes were visualized in the pleural fluid smears; this may suggest pleural effusion can be a manifestation of canine leishmaniosis. The cytological findings of the pleural fluid, in this case, were similar to the previously documented case that revealed macrophages with intracytoplasmic structures compatible with Leishmania amastigotes in pericardial fluid [8].

The cytological analysis of the pleural fluids in this case reveals typical characteristics of an exudate. The pleural fluid observed at necropsy appeared turbid, indicating a high cellular concentration, which was confirmed upon cytological evaluation after centrifugation. Additionally, the presence of necrotic and hemorrhagic thoracic lymph nodes supports the presence of an active inflammatory process, commonly associated with exudates. Exudates typically occur due to inflammatory processes that increase the permeability of pleural membranes or obstruct lymphatic flow, leading to the accumulation of fluid rich in proteins and cells. This classification is supported by Light’s criteria, which distinguish exudates from transudates based on parameters such as the pleural fluid to serum protein ratio and the lactate dehydrogenase (LDH) activity in the pleural fluid [13]. Although a complete biochemical analysis was not performed in this case, the description of the fluid and associated pathological conditions are consistent with those of an exudate.

The tests carried out for the confirmation of the diagnosis in the present case revealed that even in the face of a negative result, other exams must be performed, including cytopathology. The diagnosis of canine leishmaniosis is based on some criteria: parasitological, with detection of the parasite or its DNA—direct diagnostics and immunological tests—indirect diagnostics [14].

Despite the negative serological test result, the cytology of various lymph nodes, the bone marrow, and bloods smear of the ear tip of the animal in this study revealed macrophages containing intracytoplasmic structures consistent with Leishmania amastigotes. The direct visualization of the agent provides an immediate and conclusive diagnosis, although serological tests such as ELISA are recommended as the first approach in suspected cases [8]. Despite its high sensitivity, the ELISA test can yield false-negative results [14]. Therefore, it is recommended that cytological evaluation of the lymph nodes and bone marrow be conducted even in the presence of a negative test result.

In this study, the dog underwent an ovariohystectomy surgery, and after two months, the dog exhibited weight loss, generalized lymphadenomegaly, onychogryphosis, areas of alopecia, and pale mucous membranes. These findings are commonly observed in typical cases of canine leishmaniosis [6]. Despite the animal showing no signs of *Leishmania* infection at the time of surgery, this procedure may have resulted in a decrease in immunity and exacerbation of the infection. It is possible that an immunological abnormality occurred in this animal two months post-surgery. In humans, cases of leishmaniosis with pleural effusion in immunocompetent patients have been reported. However, this condition is more commonly seen in immunocompromised patients. In such cases, leishmaniosis manifests in a very aggressive manner and causes pleural effusion. Typically, it presents in patients with fever, cachexia, and hepatosplenomegaly [7].

The dog died suddenly during a walk. Although it exhibited signs of canine leishmaniosis, such as onychogryphosis, sudden death is not a commonly expected outcome in dogs suffering from this disease. Typical clinical manifestations of canine leishmaniosis include skin lesions, local or generalized lymphadenomegaly, a thin body condition, inappetence, lethargy, splenomegaly, polyuria and polydipsia, ocular lesions, epistaxis, lameness, vomiting, and diarrhea. While exercise intolerance may also be observed [6], sudden death has been rarely reported [8]. Exercise intolerance is also described in the literature in dogs with pleural effusion, as they also present with tachypnea, dyspnea, and cyanosis [15]. In the present case, macroscopic and histopathological evaluation of the heart and nervous system showed no lesions, although neurological, cardiorespiratory, and other atypical presentations have been described in dogs infected with *Leishmania* [16].

## 4. Conclusions

Our study showed the association of canine leishmaniosis with acute pleural effusion and sudden death in a dog. This study showed the importance of cytology as a routine clinical practice. Furthermore, this study showed post-mortem macroscopic, histopathological, immunohistochemical, and cytological examinations of the pleural fluid were essential for the conclusive diagnosis of this case.

## Figures and Tables

**Figure 1 vetsci-11-00254-f001:**
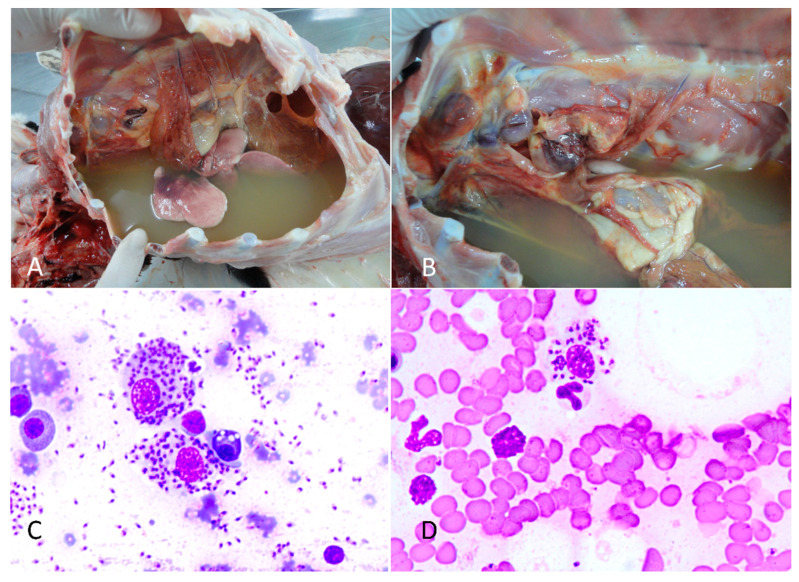
Canine Leishmaniosis associated with acute pleural effusion and sudden death in a dog. (**A**) Intrathoracic cavity filled with a large amount of yellowish liquid. (**B**) External thoracic lymph nodes showing necrosis and hemorrhage and a large amount of yellowish liquid in the chest. (**C**) Cytology of a pleural fluid smear revealed macrophages with intracytoplasmic *Leishmania* amastigotes. Rapid panoptic staining. Obj. 100×. (**D**) Cytology of a blood smear revealed macrophages with intracytoplasmic *Leishmania* amastigotes. Rapid panoptic staining. Obj. 100×.

**Figure 2 vetsci-11-00254-f002:**
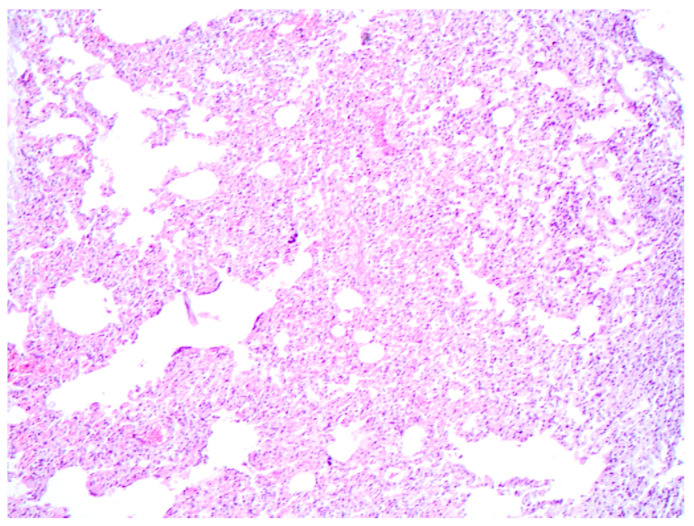
Canine Leishmaniosis associated with acute aleural effusion and sudden death in a dog. Pulmonary atelectasis characterized by collapsed alveoli, where the alveolar spaces are notably compressed. Hematoxylin and eosin. Obj. 10×.

**Figure 3 vetsci-11-00254-f003:**
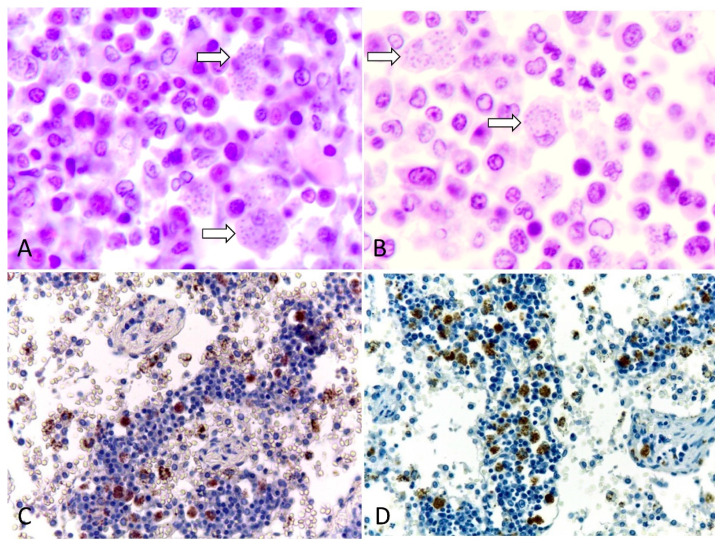
Canine Leishmaniosis associated with acute pleural effusion and sudden death in a dog. (**A**) Lymph nodes showing macrophages with intracytoplasmic *Leishmania* amastigotes (arrows). Hematoxylin and eosin. Obj. 100×. (**B**) Bone marrow showing *Leishmania* amastigotes in the cytoplasm of macrophages (arrow). Hematoxylin and eosin. Obj. 100×. (**C**) Immunohistochemistry of a tracheobronchial lymph node. Anti-leishmania, DAB chromogen, counterstained with Harris’s hematoxylin. (**D**) Immunohistochemistry of a sternal lymph node with marked positive immunostaining. Anti-Leishmania, DAB chromogen counterstained with Harris’s hematoxylin. Obj. 40×.

## Data Availability

Data contained within the Article.

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
