# Peer review of "Canine Leishmaniosis Associated with Acute Pleural Effusion and Sudden Death in a Dog"

_vetsci, 2024, doi:10.3390/vetsci11060254_

Round 1
Reviewer 1 Report
Comments and Suggestions for Authors
Overall comments
I reviewed the Manuscript (vetsci-3006546) entitled “Canine Leishmaniasis Associated with Acute Pleural Effusion and Sudden Death in a Dog” by Brito and colleagues. This study aimed to describe, for the first time, a case of acute pleural effusion as a result of leishmaniasis and necrotizing lymphadenitis in a dog.
Though of interest, the presentation of the case and its interpretation present several limitations.
Firstly, a characterisation of the pleural effusion is lacking, specifically whether it is a transudate or exudate. This characterisation is of significant importance for the identification of the potential underlying pathophysiological mechanism that may have precipitated the effusion. It also serves to reduce the range of differential diagnoses. Moreover, with the exception of the serological test for leishmaniasis, which returned a negative result, the absence of laboratory tests precludes the ability to ascertain pathological findings related to active forms of leishmaniasis. It is indubitable that direct visualisation of the parasite is of paramount importance for diagnostic purposes. However, it is erroneous to posit that the optimal site for detecting leishmania amastigotes is the ear tip; rather, it is the lymph nodes and bone marrow and cutaneous lesions.
Therefore, based on the major weaknesses outlined, I recommend that the Authors address the main points raised in detailed comments before the manuscript could be considered for publication.
Detailed comments
Page 2
2.1. Clinical Features: Were pre-operative examinations performed on the dog? Should such examinations have been carried out, it would be beneficial to provide a detailed account of the findings.
2.2. Gross Pathology and Cytology: It is stated that the cytology was prepared from the supernatant. However, this is likely a typing error, as the correct procedure is to prepare cytology from the sediment.
2.3 Histopathology: Figure 3 is missing; it may be that you are referring to Figures 2A and 2B.
Page 4
Discussion
“Aspects of pleural effusion in this animal are similar to reports described in the literature, which characterize it as the accumulation of fluid in the chest. Pleural effusion represents a disturbance of this equilibrium, because of both increased production and decreased resorption.” The meaning of this sentence is not clear. The sentence in question would benefit from reformulation and clarification in order to more effectively convey the intended concept.
“In dogs, the main causes of effusions were heart disease, neoplasms, liver disease, hypoproteinaemia, and pyometra”. For the purposes of this paper, it is only pertinent to indicate the causes of pleural effusion and not those of peritoneal effusion.
“However, in this case, no cardiac alterations were found that would justify an increase in hydrostatic pressure. Also, there were no lesions compatible with edema related to hypoproteinemia, as no hepatic lesions were found.” The classification of the effusion would facilitate the narrowing of the field of differential diagnoses.
“The renal lesions were mild to moderate. Furthermore, no other findings related to hypoproteinemia in dogs with uremia, such as pulmonary edema, were observed”. The meaning of this sentence is not clear.
“Diagnosis of canine leishmaniasis is based on some criteria.” It would be preferable to delineate these criteria in greater detail.
“In typical cases of canine Leishmaniasis, the most commonly observed physical examination findings are signs of exercise intolerance.” Although it has been reported, exercise intolerance is not among the most common clinical signs of canine leishmaniasis.
Comments on the Quality of English LanguageEnglish language deserves revisions in several parts impairing the comprehension and revision process of the text.
Author Response
Reviewer's Comment:
I reviewed the Manuscript (vetsci-3006546) entitled “Canine Leishmaniasis Associated with Acute Pleural Effusion and Sudden Death in a Dog” by Brito and colleagues. This study aimed to describe, for the first time, a case of acute pleural effusion as a result of leishmaniasis and necrotizing lymphadenitis in a dog.
Though of interest, the presentation of the case and its interpretation present several limitations.
Firstly, a characterisation of the pleural effusion is lacking, specifically whether it is a transudate or exudate. This characterisation is of significant importance for the identification of the potential underlying pathophysiological mechanism that may have precipitated the effusion. It also serves to reduce the range of differential diagnoses. Moreover, with the exception of the serological test for leishmaniasis, which returned a negative result, the absence of laboratory tests precludes the ability to ascertain pathological findings related to active forms of leishmaniasis. It is indubitable that direct visualisation of the parasite is of paramount importance for diagnostic purposes. However, it is erroneous to posit that the optimal site for detecting leishmania amastigotes is the ear tip; rather, it is the lymph nodes and bone marrow and cutaneous lesions.
Therefore, based on the major weaknesses outlined, I recommend that the Authors address the main points raised in detailed comments before the manuscript could be considered for publication.
Response:
Dear Reviewer, thank you for the comments and suggestions for improving the article. We consider all your corrections very important and have accepted all of them. You can check this in the new text of the manuscript. The effusion in this case was considered an exudate. Unfortunately, the analysis of the protein content in the effusion fluid was not performed at the time of necropsy, and this analysis can no longer be carried out. However, some aspects indicate that it is an exudative effusion, as the fluid was cloudy, while the general transudate is clear. Modified transudates can have a cloudy or hemorrhagic appearance but have low cellularity.
In this case, there was considerable inflammation and necrosis of the thoracic lymph nodes. Exudate results from inflammation and injury, while transudate is caused by unbalanced hydrostatic and osmotic pressure. Transudate is a simpler fluid, with low protein and cell content, resulting from a non-inflammatory imbalance in blood vessels. Exudate, as in this case, is a more complex fluid, with a high concentration of proteins and cells, formed in response to an inflammatory process. This information has been added to the manuscript. Regarding diagnostic methods, immunohistochemistry had confirmed the presence of the agent amid the lymph node lesions. We agree that the puncture of lymphoid organs had been performed, but this had not been reported in the previous version of the manuscript. This description has been included in the manuscript.
Detailed comments
Page 2
Reviewer's Comment:
2.1. Clinical Features: Were pre-operative examinations performed on the dog? Should such examinations have been carried out, it would be beneficial to provide a detailed account of the findings.
Response:
Pre-operative tests were conducted two months before the dog's death, after it had been rescued and undergone spay surgery. This examination included only a complete blood count. According to the owner, the values were within the normal range for the species. Unfortunately, these tests were not conducted by the authors of this article. The animal was received by us after death, for necropsy examination. The animal's owner does not have the results of these tests. However, we have added to the text a discussion on the importance of performing a wide range of tests on animals before undergoing surgery, as the surgical procedure may decrease the animal's immunity; a latent disease may worsen, leading to the death of the patient, as in this case
Reviewer's Comment:
2.2. Gross Pathology and Cytology: It is stated that the cytology was prepared from the supernatant. However, this is likely a typing error, as the correct procedure is to prepare cytology from the sediment.
Response:
Thank you. Corrected in the text. During the necropsy, a sample of the intrathoracic fluid was centrifuged, and the sediment was used for the preparation of cytological smears.
Reviewer's Comment:
2.3 Histopathology: Figure 3 is missing; it may be that you are referring to Figures 2A and 2B.
Response:
Thank you. Corrected in the text.
Page 4
Discussion
Reviewer's Comment:
“Aspects of pleural effusion in this animal are similar to reports described in the literature, which characterize it as the accumulation of fluid in the chest. Pleural effusion represents a disturbance of this equilibrium, because of both increased production and decreased resorption.” The meaning of this sentence is not clear. The sentence in question would benefit from reformulation and clarification in order to more effectively convey the intended concept.
Response:
The sentence was modified in the manuscript to ‘Pleural effusion, the abnormal accumulation of fluid within the pleural space, represents an imbalance between the production and resorption of pleural fluid. In this animal, the aspects of pleural effusion mirror those described in the literature, further solidifying the understanding of this condition.’
Reviewer's Comment:
“In dogs, the main causes of effusions were heart disease, neoplasms, liver disease, hypoproteinaemia, and pyometra”. For the purposes of this paper, it is only pertinent to indicate the causes of pleural effusion and not those of peritoneal effusion.
Response:
The sentence was modified in the manuscript to “In dogs, the main causes of pleural effusions were heart disease, neoplasms, congenital disorders, and infections of the thoracic cavity.”
Reviewer's Comment:
“However, in this case, no cardiac alterations were found that would justify an increase in hydrostatic pressure. Also, there were no lesions compatible with edema related to hypoproteinemia, as no hepatic lesions were found.” The classification of the effusion would facilitate the narrowing of the field of differential diagnoses.
Response:
The effusion in this case was considered an exudate. Unfortunately, the analysis of the protein content in the effusion fluid was not performed at the time of necropsy, and this analysis can no longer be carried out. However, some aspects indicate that it is an exudative effusion, as the fluid was cloudy, while the general transudate is clear. Modified transudates can have a cloudy or hemorrhagic appearance but have low cellularity.
In this case, there was considerable inflammation and necrosis of the thoracic lymph nodes. Exudate results from inflammation and injury, while transudate is caused by unbalanced hydrostatic and osmotic pressure. Transudate is a simpler fluid, with low protein and cell content, resulting from a non-inflammatory imbalance in blood vessels. Exudate, as in this case, is a more complex fluid, with a high concentration of proteins and cells, formed in response to an inflammatory process. This information has been added to the manuscript.
Reviewer's Comment:
“The renal lesions were mild to moderate. Furthermore, no other findings related to hypoproteinemia in dogs with uremia, such as pulmonary edema, were observed”. The meaning of this sentence is not clear.
Response:
The sentence was modified in the manuscript to “The renal lesions observed in the dog from this study ranged from mild to moderate in severity. Notably, there was an absence of other findings typically associated with hypoproteinemia in dogs with uremia, such as ascites and pulmonary edema. Despite the lack of blood tests performed on the patient, the absence of findings related to uremia in this dog suggests that the hypoproteinemia had not reached a level severe enough to cause hydrothorax. Therefore, the mild to moderate nature of the renal lesions and the lack of additional findings related to hypoproteinemia indicate that the dog's overall health was not significantly compromised by the renal disease.”
Reviewer's Comment:
“Diagnosis of canine leishmaniasis is based on some criteria.” It would be preferable to delineate these criteria in greater detail.
Response:
The diagnostic criteria were included in the manuscript text and discussed in relation to the methods used in the present study
Reviewer's Comment:
“In typical cases of canine Leishmaniasis, the most commonly observed physical examination findings are signs of exercise intolerance.” Although it has been reported, exercise intolerance is not among the most common clinical signs of canine leishmaniasis.
Response:
We agree with the reviewer's comment. Indeed, this finding in the dog from the present study highlights the importance of publishing this report so that veterinary clinicians can include this finding in their list of signs associated with canine leishmaniasis.
Reviewer's Comment:
Comments on the Quality of English Language
English language deserves revisions in several parts impairing the comprehension and revision process of the text.
Response:
We have conducted a thorough review of the English language. Furthermore, the authors have committed to the Associate Editor to submit this manuscript to a company specializing in English language review before publishing the final version of the article.

Reviewer 2 Report
Comments and Suggestions for Authors il testo con la fotocamera Thank you for considering me for this review.
Author Response
Dear Reviewer,
We appreciate your willingness to review our article.
Best Regards
Reviewer 3 Report
Comments and Suggestions for Authors
Dear Authors I reviewed your manuscript entitled "Canine Leishmaniasis Associated with Acute Pleural Effusion and Sudden Death in a Dog” and can conclude that it can be accepted, but in order to be accepted you should take care of following issues:
1) There are some wrong citations or wrong references cited that you should change with other, correct references (it is written in corrected version of the manuscript)
2) Did you stain organ smears with Giemsa? With them, you can provide beautiful pictures of amastigotes as a direct proof of leishmaniosis.
3) Please show the figure of the bloodsmear of the ear-tip that you mentioned earlier – as a proof of final diagnosis. It is mentioned in discussion and conclusion. You can not write something in conclusion without any proof of it.
4) On Figure 1C there are no visible amastigotes (the three dots present in the macrophage are not amastigotes). Please provide figure which is showing the amastigotes or don’t show it at all.
5) Please provide better resolution of Figure 2A and 2B – I can’t say the small little dots are really amastigotes. When I zoom in –it is too blurry. It could be great if you could show that on new figures shown on higher magnification.
6) Did you test the animal against other infectious diseases?
7) Did you re-test the animal with the same test and did you try to use some other test? IFAT, immunochromatographic test. It is almost impossilble for animal to have amastigotes in their bodies and not to show any detectable antibodies. In immunocompromised animals it is possible, but then you can’t say it is associated with Leishmania-in that case leishmaniosis is accidental finding.
8) Title has to be changed in “"Canine Leishmaniosis Associated with Acute Pleural Effusion and Sudden Death in a Dog”
9) You should arrange the references according to the Instructions for the authors of the Journal
10) Please write name of the genus in italic. Correct that in whole text.
11) Change all the words “leishmaniasis” into “leishmaniosis”
12) All the other corrections that should be made are in corrected version of the manuscript
Best regards

Minor editing of English language required. Almost great.
Author Response
Reviewer's Comment:
Dear Authors I reviewed your manuscript entitled "Canine Leishmaniasis Associated with Acute Pleural Effusion and Sudden Death in a Dog” and can conclude that it can be accepted, but in order to be accepted you should take care of following issues:
Response:
Dear Reviewer, thank you for your suggestions for improving the article.
Reviewer's Comment:
1) There are some wrong citations or wrong references cited that you should change with other, correct references (it is written in corrected version of the manuscript)
Response: The incorrect references have been updated.
Reviewer's Comment:
2) Did you stain organ smears with Giemsa? With them, you can provide beautiful pictures of amastigotes as a direct proof of leishmaniosis.
Response: Unfortunately, the slides were not stained with Giemsa. However, we replaced the cytological evaluation image with another one at higher magnification and greater detail of the amastigote forms of Leishmania.
Reviewer's Comment:
3) Please show the figure of the bloods mear of the ear-tip that you mentioned earlier – as a proof of final diagnosis. It is mentioned in discussion and conclusion. You can not write something in conclusion without any proof of it.
Response:
The figure of the blood smear from the ear-tip has been included.
Reviewer's Comment:
4) On Figure 1C there are no visible amastigotes (the three dots present in the macrophage are not amastigotes). Please provide figure which is showing the amastigotes or don’t show it at all.
Response: we replaced the cytological evaluation image with another one at higher magnification and greater detail of the amastigote forms of Leishmania.
Reviewer's Comment:
5) Please provide better resolution of Figure 2A and 2B – I can’t say the small little dots are really amastigotes. When I zoom in –it is too blurry. It could be great if you could show that on new figures shown on higher magnification.
Response:
We have included additional images showing the amastigote forms of Leishmania in lymph nodes and bone marrow, both at 100x magnification. However, these organs exhibited severe inflammation, which complicates the visualization of parasitic structures in histopathology, unlike the considerable detail that is facilitated in cytological evaluation at the same magnification.
Reviewer's Comment:
6) Did you test the animal against other infectious diseases?
Response:
During the necropsy, various samples were collected and preserved frozen for additional diagnostic purposes. However, cytological evaluation of blood smears and lymphoid organs did not reveal the presence of other agents. Histopathological examination of all organs was also conducted, and no evidence of other infectious agents was found. Consequently, no further tests or examinations were performed.
Reviewer's Comment:
7) Did you re-test the animal with the same test and did you try to use some other test? IFAT, immunochromatographic test. It is almost impossilble for animal to have amastigotes in their bodies and not to show any detectable antibodies. In immunocompromised animals it is possible, but then you can’t say it is associated with Leishmania-in that case leishmaniosis is accidental finding.
Response:
In our routine diagnostic experience, we have observed some false-negative cases when tested with the EIE-LVC kit produced by BioManguinhos/FIOCRUZ. Moreover, studies have also shown that serological tests such as the indirect immunofluorescent assay (IFA), the direct agglutination test (DAT), and the enzyme-linked immunosorbent assay (ELISA) have shown variable results in detecting Leishmania. Unfortunately, this dog was abandoned and rescued, but the owner did not have the financial resources for additional testing. Furthermore, the dog was sent to us only after its death, for necropsy, which prevented a thorough clinical investigation of this case.
Reviewer's Comment:
8) Title has to be changed in “"Canine Leishmaniosis Associated with Acute Pleural Effusion and Sudden Death in a Dog”
Response:
This has been corrected as recommended.
9) You should arrange the references according to the Instructions for the authors of the Journal
Response:
This has been corrected as recommended.
10) Please write name of the genus in italic. Correct that in whole text.
Response:
This has been corrected as recommended.
11) Change all the words “leishmaniasis” into “leishmaniosis”.
Response:
This has been corrected as recommended.
12) All the other corrections that should be made are in corrected version of the manuscript.
Response:
This has been corrected as recommended.

Reviewer 4 Report
Comments and Suggestions for Authors
This case report presents interesting and orginal findings of canine leishmaniasis associated with acute pleural effusion and sudden death in a dog. I have some minor considerations (see the comments below).
Leishmania amastigotes - replace with Leishmania amastigotes, everywhere in text.
Keywords: “Cur” - please replace this word with a more appropriate and familiar one for readers.
Introduction
Only two cases of acute effusion have been reported in dogs affected by Leishmania sp., involving pericardial effusion [7] - you stated one case (7) instead of two, please complete.
Case presentation
Laurenti et al. [8] - refers to reference [9], please correct.
Discussion
“Diagnosis of canine leishmaniasis is based on some criteria [11] “- the sentence seems incomplete, please expand it.
Change this long sentence to make it clearer : “The animal of this study, although the serological test revealed a negative result, in the blood smear made from a sample obtained from an ear-tip revealed macrophages with intracytoplasmic structures compatible with Leishmania amastigotes” in e.g.
“Despite the negative serological test result, the blood smear from an ear-tip of the animal in this study showed macrophages containing intracytoplasmic structures consistent with Leishmania amastigotes” and reshape the second part (“showing that direct..”) of the sentence into a new sentence.
“..animal underwent a surgical procedure” - Please specify which surgical procedure.
Comments on the Quality of English Language
Some sentences should be styled.
Author Response
Reviewer's Comment:
This case report presents interesting and orginal findings of canine leishmaniasis associated with acute pleural effusion and sudden death in a dog. I have some minor considerations (see the comments below).
Response:
Dear Reviewer,
thank you for the comments and suggestions for improving the article. We consider all your corrections very important and have accepted all of them. You can check this in the new text of the manuscript.
Reviewer's Comment:
Leishmania amastigotes - replace with Leishmania amastigotes, everywhere in text.
Response:
This has been corrected as recommended.
Reviewer's Comment:
Keywords: “Cur” - please replace this word with a more appropriate and familiar one for readers.
Response:
This has been corrected as recommended.
Introduction
Reviewer's Comment:
Only two cases of acute effusion have been reported in dogs affected by Leishmania sp., involving pericardial effusion [7] - you stated one case (7) instead of two, please complete.
Response:
This has been corrected as recommended.
Case presentation
Reviewer's Comment:
Laurenti et al. [8] - refers to reference [9], please correct.
Response:
This has been corrected as recommended.
Discussion
“Diagnosis of canine leishmaniasis is based on some criteria [11] “- the sentence seems incomplete, please expand it.
Response:
This has been corrected as recommended: “Diagnosis of canine leishmaniosis is based on some criteria: parasitological, with detection of the parasite or its DNA – direct diagnostics and immunological tests – indirect diagnostics.”
Reviewer's Comment:
Change this long sentence to make it clearer : “The animal of this study, although the serological test revealed a negative result, in the blood smear made from a sample obtained from an ear-tip revealed macrophages with intracytoplasmic structures compatible with Leishmania amastigotes” in e.g.
“Despite the negative serological test result, the blood smear from an ear-tip of the animal in this study showed macrophages containing intracytoplasmic structures consistent with Leishmania amastigotes” and reshape the second part (“showing that direct..”) of the sentence into a new sentence.
Response:
This has been corrected as recommended.
Reviewer's Comment:
“..animal underwent a surgical procedure” - Please specify which surgical procedure.
Response:
This has been corrected as recommended: “the dog underwent ovariohystectomy surgery”
Comments on the Quality of English Language
Reviewer's Comment:
Some sentences should be styled.
Response:
We have conducted a thorough review of the English language. Furthermore, the authors have committed to the Associate Editor to submit this manuscript to a company specializing in English language review before publishing the final version of the article.

Round 2
Reviewer 1 Report
Comments and Suggestions for Authors
Overall comments
I have read the revised version of the manuscript (vetsci-3006546) entitled “Canine Leishmaniasis Associated with Acute Pleural Effusion and Sudden Death in a Dog” by Brito and colleagues. The text has been amended accordingly to the reviewer comments. Therefore, in my opinion, the manuscript is suitable for publication in Veterinary Sciences pending some minor revisions.
Minor comments
Line 30: “to the authors' knowledge, this association has not yet been described.” specify 'in dogs'
Line 129: correct caption by indicating it as figure 2 and not figure 3
Line 154: correct caption by indicating it as figure 3 and not figure 2
Line 157: “Immunohistochemistry of a tracheobronchial lymph node. Anti-leishmania, DAB chromogen, counterstained with Harris's hematoxylin” should refer to figure D but is not specified
Line 183-188: In renal disease, hypoproteinaemia is associated with protein-losing nephropathy, rather than uremia. Nevertheless, in the context of severe hypoproteinaemia, it is anticipated that a transudate would be observed, rather than an exudate.
Comments on the Quality of English LanguageThe quality of English language is sufficiently clear.
Author Response
Reviewer's Comment:
I have read the revised version of the manuscript (vetsci-3006546) entitled “Canine Leishmaniasis Associated with Acute Pleural Effusion and Sudden Death in a Dog” by Brito and colleagues. The text has been amended accordingly to the reviewer comments. Therefore, in my opinion, the manuscript is suitable for publication in Veterinary Sciences pending some minor revisions:
Response:
Dear Reviewer, thank you for your suggestions for improving the article. We have accepted all suggestions.
Reviewer's Comment:
Line 30: “to the authors' knowledge, this association has not yet been described.” specify 'in dogs'
Response:
This has been changed as recommended.
Reviewer's Comment:
Line 129: correct caption by indicating it as figure 2 and not figure 3.
Response:
This has been changed as recommended.
Reviewer's Comment:
Line 154: correct caption by indicating it as figure 3 and not figure 2.
Response:
This has been changed as recommended.
Reviewer's Comment:
Line 157: “Immunohistochemistry of a tracheobronchial lymph node. Anti-leishmania, DAB chromogen, counterstained with Harris's hematoxylin” should refer to figure D but is not specified.
Response:
This has been changed as recommended.
Reviewer's Comment:
Line 183-188: In renal disease, hypoproteinaemia is associated with protein-losing nephropathy, rather than uremia. Nevertheless, in the context of severe hypoproteinaemia, it is anticipated that a transudate would be observed, rather than an exudate.
Response:
Unfortunately, we were not able to follow this animal while it was still alive; we only came into contact with the dog at the time of necropsy. Therefore, we were not able to assess hypoproteinemia. However, we agree with the comment about protein loss in nephropathy. Therefore, we have revised the text to: “The renal injuries observed in the dog in this study ranged from mild to moderate in severity. Notably, there was an absence of other findings typically associated with hypoproteinemia in dogs with uremia, such as ascites and pulmonary edema [5]. However, despite the lack of blood tests performed on the patient, given the renal injuries observed, although of low severity, the liquid present in the thoracic cavity of this dog could also be of the transudate type, due to hypoproteinemia. Therefore, the liquid may be a mixture of exudate and transudate, as there was marked cellularity in the sediment, but a hydrothorax evidenced by hypoproteinemia cannot be ruled out [5].”

Reviewer 3 Report
Comments and Suggestions for Authors
Dear Authors, everything's corrected and my opinion is that now your manuscript should be accepted as it is. Congrats! :-)
Author Response
Dear Reviewer, thank you for your suggestions for improving the article.